# Enhancing Uncertainty-Based Hallucination Detection with Stronger Focus

**Tianhang Zhang[1], Lin Qiu[2], Qipeng Guo[2], Cheng Deng[1], Yue Zhang[3]**
**Zheng Zhang[2], Chenghu Zhou[4], Xinbing Wang[1] and Luoyi Fu[1]**
[1]Shanghai Jiaotong University, China    [2]Amazon AWS AI
[3]Westlake University, China    [4]IGSNRR, Chinese Academy of Sciences, China
{zhangtianhang, davendw, xwang8, yiluofu}@sjtu.edu.cn
{quln, gqipeng, zhaz}@amazon.com, yue.zhang@wias.org.cn

## Abstract

Large Language Models (LLMs) have gained significant popularity for their impressive performance across diverse fields. However, LLMs are prone to hallucinate untruthful or nonsensical outputs that fail to meet user expectations in many real-world applications. Existing works for detecting hallucinations in LLMs either rely on external knowledge for reference retrieval or require sampling multiple responses from the LLM for consistency verification, making these methods costly and inefficient. In this paper, we propose a novel reference-free, uncertainty-based method for detecting hallucinations in LLMs. Our approach imitates human focus in factuality checking from three aspects: 1) focus on the most informative and important keywords in the given text; 2) focus on the unreliable tokens in historical context which may lead to a cascade of hallucinations; and 3) focus on the token properties such as token type and token frequency. Experimental results on relevant datasets demonstrate the effectiveness of our proposed method, which achieves state-of-the-art performance across all the evaluation metrics and eliminates the need for additional information.[1]

## 1 Introduction

Large Language Models (LLMs) have garnered substantial attention for their remarkable performance across various domains, such as finance (Wu et al., 2023; Lopez-Lira and Tang, 2023), medicine (Javaid et al., 2023; Lee et al., 2023), and education (Tlili et al., 2023; Baidoo-Anu and Owusu Ansah, 2023). These models exhibit an extraordinary capacity to generate natural language texts with high levels of coherence, fluency, and informativeness. Nevertheless, a significant obstacle confronting LLMs is the risk of producing hallucinations (Shen et al., 2023b; Sallam,

---

[1]Code can be found at https://github.com/zthang/focus.

2023), which refers to the generated text that is untruthful or nonsensical (Ji et al., 2023; Bang et al., 2023). Hallucinations are a common occurrence in almost all applications (Xu et al., 2023), thus undermining the reliability and trustworthiness of LLMs, especially in scenarios where accuracy and veracity are essential.

Existing studies on hallucination detection for LLMs can be broadly divided into two categories: (i) retrieval-based methods (Min et al., 2023; Liu et al., 2023), which evaluate the veracity of the given text against knowledge bases, and (ii) sampling-based methods (Mündler et al., 2023; Manakul et al., 2023), which assess information consistency between the evaluated text and additional sampled responses from the same LLM. However, retrieval-based methods depend heavily on the external knowledge that may not always be accessible. And sampling-based methods require multiple responses from the LLM for the information consistency verification or model training, making these methods costly and inefficient.

To address the above issues, we propose a novel reference-free, uncertainty-based method to detect hallucinations in LLMs that are factually incorrect according to the world knowledge. The proposed method relies exclusively on the LLM output text, thereby eliminating the need for additional resources like sampled responses from LLM or external knowledge, as well as further training based on such data. Our basic idea is to use a proxy language model for calculating the probability of each token in a given text. Based on the calculated probability, we can compute the hallucination score at both token and sentence level using uncertainty-based metrics (Guerreiro et al., 2022; Xiao and Wang, 2021), where tokens and sentences with high hallucination scores are identified as candidate hallucinated contents. Our assumption is that a powerful enough LLM should assign a low probability to tokens that make up hallucinated information, since

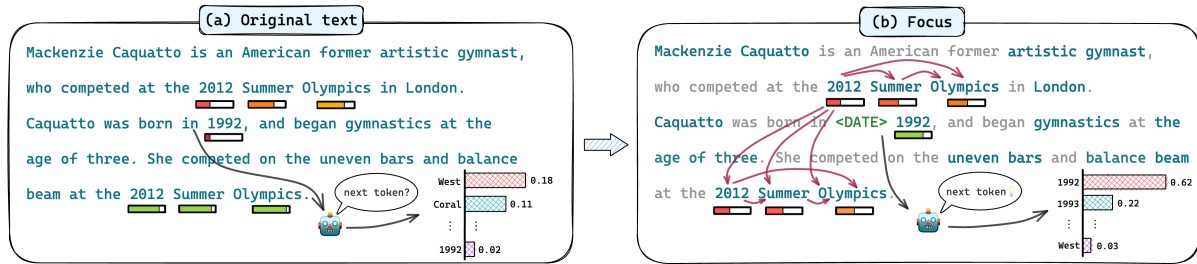

Figure 1: (a) Using a naive proxy model can hinder the focus on hallucination itself: 1) considering all tokens within the given text may introduce noise; 2) the hallucinated tokens might be assigned high probabilities (green bar) due to the overconfidence problem; 3) factual tokens may receive low probabilities (red bar) due to the underconfidence problem. (b) To strengthen such focus, we imitate how humans perform factuality checking from three aspects: 1) focus on the informative keywords; 2) focus on the preceding words by propagating the uncertainty through attention weights; 3) focus on the token properties by providing entity type before each named entity.

they deviate from the world knowledge the model has learned during its training stage.

The above method serves as a base framework, which can be limited by the inherent characteristics of the prediction probability from a naive proxy model. Such a model functions as a general probability estimator, its predictions reflect syntactic, semantic and other sources of information, which can hinder the focus on hallucination itself as illustrated in Figure 1a.

Firstly, the proxy model ignores varying degrees of informativeness, which may introduce noise. Secondly, the probabilities assigned by LMs are general and can deviate from factuality confidence in different contexts. For instance, the proxy model can be **overconfident** if the historical context contains surface tokens that are correlated with a hallucinated token, or the historical context features exposure bias (Bengio et al., 2015; Iqbal and Qureshi, 2022) due to the auto-regressive nature of generative process. One example is shown in Figure 1a, where hallucinated tokens "2012 Summer Olympics" are assigned high probabilities. In addition, a proxy model can be **underconfident** if there are many plausible choices of topic directions to continue a context, despite that the hallucination involves different tokens within only one topic direction. One example is shown in Figure 1a, where the factual token "1992" received a low probability due to the competitors like "West" and "Coral".

To strengthen the focus on hallucination, we take inspiration from human factuality checking, which can include at least three specific considerations as depicted in Figure 1b:

- **Focus on the informative keywords**: the keywords that express salient information will be extracted for the calculation of hallucination scores at both sentence-level and passage-level.

- **Focus on the preceding words**: we propagate the uncertainties of previous tokens to the subsequent ones according to their attention weights to alleviate the overconfidence problem. This approach is based on the hypothesis that words that are strongly connected to unreliable tokens may also be influenced by these inaccuracies, which can trigger a chain reaction of hallucinations.

- **Focus on the token properties**: the predicted token probability is conditioned on its entity type (if any) and adjusted by its inverse document frequency (IDF). This results in a probability distribution that aligns more closely with human evaluation in a posterior manner, thus mitigating the underconfidence problem.

In summary, our primary contribution is that we introduce a novel reference-free, uncertainty-based approach for detecting hallucinations in LLMs. Our approach does not require additional sampled responses or external knowledge bases, making it simple and cost-effective. Experimental results demonstrate that our proposed method achieves state-of-the-art performance on the WikiBio GPT-3 dataset across various models with different scales, and shows effectiveness in detecting hallucinations within summaries generated by small models.

## 2 Related Work

### 2.1 Hallucinations in Text Generation

Hallucinations are prevalent phenomenon in deep learning-based models employed for various text generation tasks (Xu et al., 2023), such as abstractive summarization (Huang et al., 2021; Nan et al., 2021), dialogue generation (Dziri et al.,

2022; Rashkin et al., 2021) and question answering (Longpre et al., 2021; Su et al., 2022). Hallucinations present significant challenges in text generation tasks, as they can lead to inaccurate or misleading results, which is unacceptable in most user-oriented applications (Liu et al., 2022; Xu et al., 2023; Rebuffel et al., 2022).

## 2.2 Hallucination Detection

Previous studies on hallucination detection have primarily concentrated on identifying hallucinations produced by small models (fewer than 1b parameters) that are tailored for specific tasks. For instance, Kasner et al. (2021) combined a rule-based system and a pretrained language model to identify hallucinations in table-to-text generation. Guerreiro et al. (2022) adopted the average log-probability across all the tokens in the output sequence as the model uncertainty metric for detecting hallucinations in machine translation. Dale et al. (2022) attempted to detect hallucinations by evaluating the percentage of the source contribution to the generated text. However, the hallucination patterns exhibited by LLMs tend to be divergent from those in small models (Guerreiro et al., 2023), posing challenges for the generalization of these methods on detecting hallucinations in LLMs. Accordingly, hallucination detection in small models is not within the primary scope of this paper.

The widespread incorporation of LLMs across a diverse range of applications has drawn substantial attention from researchers towards the issue of hallucinations within LLMs (Bang et al., 2023; Shen et al., 2023a; Alkaissi and McFarlane, 2023). For instance, Min et al. (2023) introduced FACTSCORE to evaluate the correctness of each atomic fact in the generated text by referencing a knowledge source. Mündler et al. (2023) aimed to detect hallucinations by examining whether two sampled sentences generated at the same position within a text contradict each other. A recent work by Manakul et al. (2023) proposed SelfCheckGPT, a black-box approach for detecting hallucinations in LLM-generated responses. The primary premise of SelfCheckGPT is that when the LLM is uncertain about a given concept, the sampled responses may contain inconsistent facts. Nonetheless, these methods either rely on external knowledge bases or multiple responses sampled from LLM, which are resource-intensive and inefficient.

## 3 Methods

A proxy model is utilized in our method for uncertainty assessment in cases where token-level probabilities are inaccessible, such as GPT-3 (Ouyang et al., 2022). Although previous work by Manakul et al. (2023) has demonstrated the ineffective performance of using a proxy model, we attribute it to the uncertainty metrics employed. These metrics, such as the average entropy and average loss for all tokens in the sentence, are insufficiently aligned with human evaluation. We believe this issue stems from the inherent disparities in how models and humans perceive and assess information, thus limiting the capability of the uncertainty-based approach for hallucination detection.

To mitigate this problem, we imitate how humans perform factuality checking from three aspects, which will be discussed in following sections.

## 3.1 Keywords selection

Prior works (Pagnoni et al., 2021; Kryściński et al., 2020) suggest that entities are the most frequently hallucinated words in text generation. This aligns with the intuition that, when evaluating the veracity of generated results, our primary focus lies on keywords that convey the most crucial information. In this regard, we only focus on keywords identified by Spacy (Honnibal and Montani, 2017) when calculating the hallucination score at both sentence-level and passage-level. The keywords identified by Spacy can be classified into two groups. The first group comprises 18 distinct types of named entities, including person, location, date, event, organization, and others. The second group encompasses nouns that do not belong to the first group.

Specifically, for a given text $r$, we will compute a hallucination score $h_i$ for the $i$-th token $t_i$ in $r$. To fully utilize both local and global uncertainty information, $h_i$ is the sum of the negative log probability and entropy when generating $t_i$:

$$h_i = -log(p_i(t_i)) + \mathcal{H}_i \qquad (1)$$

$$\mathcal{H}_i = 2^{-\sum_{v \in \mathcal{V}} p_i(v)*log_2(p_i(v))} \qquad (2)$$

where $p_i(v)$ denotes the probability of generating the token $v$ over all tokens in the vocabulary $\mathcal{V}$ at position $i$. The hallucination score $h^s$ for the sentence $s$ is calculated by a weighted sum, where the

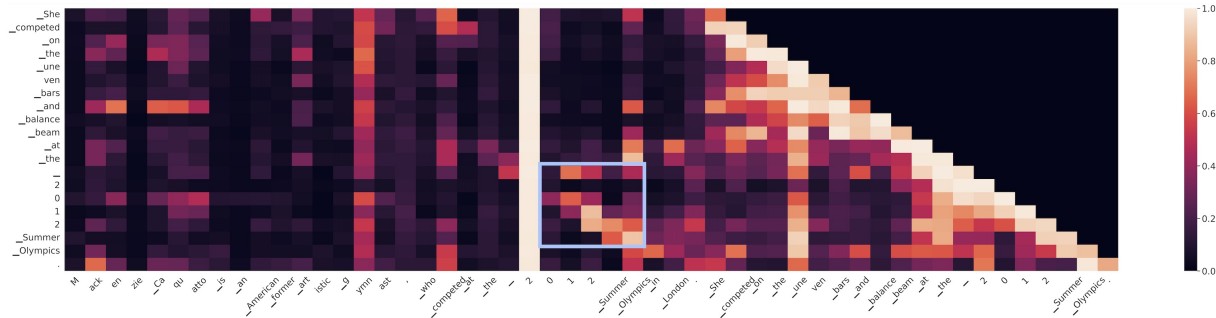

Figure 2: The attention heat map after max-pooling for all the layers and attention heads when generating the example using llama-30b, where the x-axis only presents the first and last sentence, while the y-axis only includes the last sentence due to space constraints. The brightness of each rectangle represents the attention score between the corresponding tokens, with brighter shades indicating higher scores.

weight is determined by whether $t_i$ is a keyword:

$$h^s = \frac{1}{\sum_{i=0}^{|s|-1} \mathbb{I}(t_i \in \mathcal{K})} \sum_{i=0}^{|s|-1} \mathbb{I}(t_i \in \mathcal{K}) * h_i \quad (3)$$

where $|s|$ is the number of tokens in $s$, $\mathcal{K}$ denotes the set of keywords, $\mathbb{I}(\cdot)$ is an indicator function. Moreover, this formulation can be extended to compute the passage-level hallucination score by averaging hallucination scores of keywords in the given passage.

### 3.2 Hallucination propagation

Several studies (Guerreiro et al., 2022; Xiao and Wang, 2021) have utilized token probability as a measure for hallucination detection. However, probabilities derived from a language model may not accurately reflect the factuality confidence in the generated content. Some hallucinated tokens can be assigned high probabilities when the history context contains hallucinated information, which we term as the overconfidence issue. This issue is exacerbated by the self-attention mechanism that is commonly used in transformer-based LLMs, since it introduces exposure bias (Bengio et al., 2015; Iqbal and Qureshi, 2022), which refers to the discrepancy between training and inference caused by the use of teacher forcing during the training stage. Consequently, the generated text is accepted as factual claims, even though it may be non-factual.

Figure 2 provides an example that illustrates the overconfidence issue. Considering the following text: *"Mackenzie Caquatto is an American former artistic gymnast, who competed at the 2012 Summer Olympics in London. Caquatto was born in 1992, and began gymnastics at the age of three. She competed on the uneven bars and balance beam at the 2012 Summer Olympics."* Notably, the term

"2012" makes two appearances, with the probability of its initial occurrence being significantly lower than the probability of its subsequent appearance. The visualized self-attention matrix reveals that considerable attention is given to the same phrase in the first sentence (circled with a blue box) when generating "2012 Summer Olympics" in the last sentence. However, the claim "Mackenzie Caquatto competed at the 2012 Summer Olympics in London" is untruthful.

This observation inspired us to introduce a "penalty" for tokens generated with attentions paid to unreliable tokens. In other words, we consider the hallucination scores of preceding tokens and apply them as a penalty to the current token based on their respective attention weights. Here, we only consider propagation between keywords. Specifically, we first check if the current token is a keyword as described in Section 3.1. If not, the penalty is set to zero. If it is a keyword, we normalize the attention weights between the current token and all previous keywords to obtain a penalty weight. The penalty for the current token is computed as a weighted sum of the hallucination scores associated with the previous tokens. Since the penalty can be transmitted to all the subsequent tokens via multi-hop, a coefficient $\gamma \in [0, 1]$ is introduced to ensure that the penalty diminishes geometrically with the increase in the number of hops.

Let $\hat{h}_i$ represent the hallucination score of the $i$-th token $t_i$ with an accumulated penalty, the calculation of $\hat{h}_i$ can be expressed as follows:

$$\hat{h}_i = h_i + \mathbb{I}(t_i \in \mathcal{K}) * \gamma * p_i \quad (4)$$

$$p_i = \sum_{j=0}^{i-1} w_{i,j} * \hat{h}_j \quad (5)$$

$$w_{i,j} = \frac{\mathbb{I}(t_i \in \mathcal{K}) * att_{i,j}}{\sum_{k=0}^{i-1} \mathbb{I}(t_i \in \mathcal{K}) * att_{i,k}} \quad (6)$$

where $p_i$ represents the penalty of the $i$-th token, $att_{i,j}$ denotes the attention weight between $t_i$ and $t_j$ after max-pooling for all the layers and attention heads.

### 3.3 Probability correction

Apart from the overconfidence issue, there are also instances where the model exhibits underconfidence, which can also lead to deviations in token probability from factuality confidence. We believe such underconfidence is related to the token properties, including the entity type and token frequency. As shown in Figure 1a, when generating the subsequent words following "Caquatto was born in". The model may have multiple possible choices of topic directions such as "West chester", "Coral Springs", "1992" et al, despite that the hallucination involves different tokens within a specific topic direction. Consequently, the probability of generating the date "1992" would be relatively low, given the existence of several other viable options.

This highlights the stark disparities in how models and humans assess information: when evaluating the plausibility of "1992", the model focuses meticulously on all the feasible choices with different entity types. In contrast, humans tend to intuitively include it within a tailored set of candidate words that predominantly consists of terms related to dates. Suppose there are $n$ tokens $t_{0:n-1} = t_0, t_1, ..., t_{n-1}$ in a model response $r$. Let $c(t_{0:i})$ denote the set of ideal candidate words for $t_i$ given the first $i + 1$ tokens. According to the Bayes rule, the probability of generating $t_i$ given $t_{0:i-1}$ and the candidate set can be expressed as:

$$
\begin{aligned}
&p(t_i|t_{0:i-1}, c(t_{0:i})) \\
&= \frac{p(c(t_{0:i})|t_{0:i-1}, t_i) * p(t_i|t_{0:i-1})}{p(c(t_{0:i})|t_{0:i-1})} \\
&= \frac{p(t_i|t_{0:i-1})}{p(c(t_{0:i})|t_{0:i-1})} \quad (7) \\
&= \frac{p(t_i|t_{0:i-1})}{\sum_{v \in c(t_{0:i})} p(v|t_{0:i-1})}
\end{aligned}
$$

It suggests that when assessing the rationality of a given word, the focus should be directed towards similar words rather than encompassing all possible choices. However, constructing such a candidate set poses a challenge during the model generation

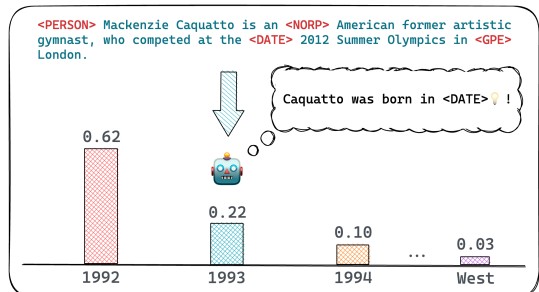

Figure 3: An example of providing entity type preceding named entities: Top-3 words that follow the incomplete sentence are all related to dates. Despite having the highest probability in Figure 1a, the token "West" is generated with a relatively low probability of 0.03.

stage, given all words are tokenized into sentence pieces. To tackle this problem, we leverage the in-context learning capability of the proxy model by inserting the entity type[2] preceding every named entity identified by Spacy as shown in Figure 3. The entity type serves as a constraint in generation, thereby enabling us to approximate the ideal candidate set $c(t_{0:i})$ in Equation 7 using tokens with a generation probability greater than a threshold $\rho$. Accordingly, the token probability distribution is corrected to assign higher probability to tokens adhering to the given entity type.

Additionally, as outlined in previous studies (Raunak et al., 2020; van der Poel et al., 2022; Demeter et al., 2020), tokens with low frequency are likely to receive lower prediction probabilities, potentially leading to the underconfidence in the model. To mitigate this issue, the probability of token $t$ is further corrected by its token IDF:

$$\hat{p}(t) = \frac{\tilde{p}(t) * idf(t)}{\sum_{v \in \mathcal{V}} \tilde{p}(v) * idf(v)} \quad (8)$$

where $\tilde{p}(t)$ denotes the probability of token $t$ across all tokens in the vocabulary $\mathcal{V}$ with entity type provided. The token IDF is calculated based on 1M documents sampled from RedPajama dataset[3].

### 3.4 Putting things together

To combine all the methods proposed above, we replace the token probability in Equation 1 and Equation 2 with $\hat{p}(t)$. Subsequently, we apply hallucination propagation to obtain the token-level hallucination score with penalty accumulated. The sentence-

---

[2]The entity type inserted in the text do not participate in hallucination propagation or hallucination score calculation.

[3]https://huggingface.co/datasets/togethercomputer/RedPajama-Data-1T-Sample

level and passage-level hallucination scores are calculated based on Equation 3.

## 4 Experiments and Results

### 4.1 Experiment setting

**Dataset**. We evaluated our proposed method on WikiBio GPT-3 dataset (Manakul et al., 2023), which, to the best of our knowledge, is the only publicly accessible dataset for LLM hallucination detection at present. Additionally, to assess the extent to which our proposed method can be applied for detecting hallucinations produced by different models, and in particular small models, we conducted supplementary experiments on the XSum-Faith (Maynez et al., 2020) and FRANK (Pagnoni et al., 2021) datasets. Given the primary focus of this paper is hallucination detection in LLM as discussed in Section 2.2, the details and results of the two datasets are provided in Appendix A.

The WikiBio GPT-3 dataset comprises 1908 annotated sentences from 238 Wikipedia passages generated by text-davinci-003. Each sentence is assigned one of the following labels: 1) *major inaccurate*, if the sentence is irrelevant to the given topic; 2) *minor inaccurate*, if the sentence includes non-factual information verifiable via web search; 3) *accurate*, if the sentence does not contain any hallucination. We provided some examples from the dataset in Appendix D. The dataset also included 20 stochastically-sampled responses from text-davinci-003 for each passage, but these were not utilized in our experiments as our method does not necessitate additional sampled responses.

In accordance with the setting in Manakul et al. (2023), sentences labeled as major inaccurate and minor inaccurate are grouped into the non-factual class, while remaining sentences are grouped into the factual class. For the non-factual* class, we first remove passages where all sentences are labeled as major inaccurate. Then, we classify remaining major inaccurate sentences as non-factual*.

**Baselines**. (i) GPT-3 (Ouyang et al., 2022) Uncertainties: GPT-3 (text-davinci-003) API returns top-5 probabilities for each generated token, which can be used to quantify its uncertainty using negative log probability and entropy. (ii) SelfCheckGPT: SelfCheckGPT (Manakul et al., 2023) is a black-box method for detecting hallucinations in LLMs, which demands additional responses sampled from the same LLM for the consistency verification.

**Metrics**. To ensure a fair comparison, we adopt

same metrics employed by SelfCheckGPT. The Area Under the Precision-Recall Curve (AUC-PR) is used to measure the performance of sentence-level hallucination detection, while the Pearson and Spearman's correlation coefficient are applied to evaluate the agreement between the passage-level hallucination score and human judgement. For space saving, AUC-PR of non-factual class is abbreviated as NonFact or NoFac in the following sections, and similarly for the other classes.

**Proxy model**. To demonstrate the generalizability of our proposed method across different scales of LLMs, we conduct experiments on **22** diverse proxy models. The specific details of these proxy models can be found in Appendix E.

**Prompts**. In experiments where entity types are not provided, we use the prompt "`This is a passage from Wikipedia about {concept}:`". Conversely, when entity types are inserted before named entities, the prompt is "`Please complete the passage below using appropriate words that follow to the given type with < > wrapped. This is a passage from Wikipedia about {concept}:`".

### 4.2 Main results

The performance comparison between our proposed method and the baseline approaches is presented in Table 1. Due to space limitations, we only display the results of LLaMA family. Comprehensive comparison results for all proxy models can be found in the Appendix H. The hyperparameters $\gamma$ and $\rho$ are set to 0.9 and 0.01, respectively. The baseline results are referenced from Manakul et al. (2023). Other implementation details can be found in Appendix B. Our key findings are as follows:

**Proxy model surpasses all the baselines**. Leveraging three proposed focus mechanisms, LLaMA-30b consistently outperforms SelfCheckGPT-Combination[4] and other baselines across all five metrics. Significantly, this is achieved without resorting to sampled responses or further training, exhibiting superior efficiency compared to Self-CheckGPT. As presented in Table 1, the performance of LLaMA family improves as the model size increases. However, this improvement is not

---

[4]We noticed that on June 11th, the authors of SelfCheck-GPT updated their results on GitHub (but not for their arXiv paper). The new approach entails a large number of ChatGPT queries for text inconsistency assessment. We do not include the results in this paper since they are contemporaneous with our work, as well as the comparison is not fair.

| Method | Sentence-level Metrics | | | Passage-level Metrics | |
|---|---|---|---|---|---|
| | NonFact | NonFact* | Factual | Pearson | Spearman |
| GPT-3 Uncertainties | | | | | |
| Avg($-\log p$) | 83.21 | 38.89 | 53.97 | 57.04 | 53.93 |
| Avg($\mathcal{H}$) | 80.73 | 37.09 | 52.07 | 55.52 | 50.87 |
| Max($-\log p$) | 87.51 | 35.88 | 50.46 | 57.83 | 55.69 |
| Max($\mathcal{H}$) | 85.75 | 32.43 | 50.27 | 52.48 | 49.55 |
| SelfCheckGPT | | | | | |
| BERTScore | 81.96 | 45.96 | 44.23 | 58.18 | 55.90 |
| QA | 84.26 | 40.06 | 48.14 | 61.07 | 59.29 |
| Unigram (max) | 85.63 | 41.04 | 58.47 | 64.71 | 64.91 |
| Combination | 87.33 | 44.37 | 61.83 | 69.05 | 67.77 |
| **Ours** | | | | | |
| LLaMA-7B$_{focus}$ | 84.26 | 40.20 | 57.04 | 64.47 | 54.73 |
| LLaMA-13B$_{focus}$ | 87.90 | 43.84 | 62.46 | 70.62 | 63.03 |
| LLaMA-30B$_{focus}$ | 89.79 | **48.80** | **65.69** | **77.15** | **73.24** |
| LLaMA-65B$_{focus}$ | **89.94** | 48.69 | 64.90 | 76.80 | 73.01 |

Table 1: Performance comparison between proposed method and baseline methods. AUC-PR is adopted as the performance metric for sentence-level hallucination detection. Passage-level performances are measured by Pearson correlation coefficient and Spearman's correlation coefficient with respect to human annotations. Results of GPT-3 and SelfCheckGPT are referenced from the paper (Manakul et al., 2023).

linearly correlated to the model size as shown in Figure 9 of Appendix F. LLaMA-65b even exhibits slightly inferior performance compared to LLaMA-30b in four of the five metrics.

Moreover, the comprehensive results across 22 proxy models as demonstrated in Table 8 affirm that within the same model family, models with more parameters tend to perform better. This can be attributed to their broader and more accurate understanding of world knowledge. In addition, when comparing different model families, models that exhibit superior performance on general NLP tasks often perform well on the WikiBio GPT-3 dataset. These observations provide valuable insights for future exploration and enhancement of our hallucination detection method.

**Focus allows small-scale models to achieve comparable performance to GPT-3**. As shown in Table 1, LLaMA-7b achieves comparable or even superior performance when compared with GPT-3 uncertainties. This observation suggests that despite being a powerful LLM with 175b parameters, GPT-3 may be similarly plagued by issues of overconfidence and underconfidence. However, neither the attention weights nor the full probability distribution of GPT-3 are accessible, otherwise, the incorporation of focus would enable uncertainties of GPT-3 to yield considerably enhanced results.

### 4.3 Analysis

Table 2 presents the results of our ablation study conducted on LLaMA-30b. The average hallucination score in Equation 1 without any proposed

| Method | NoFac | NoFac* | Fact | Pear. | Spear. |
|---|---|---|---|---|---|
| avg($h$) | 82.07 | 41.47 | 47.22 | 51.03 | 37.29 |
| +keyword | 83.01 | 41.57 | 45.82 | 56.07 | 44.77 |
| +penalty | 86.68 | 45.27 | 54.93 | 59.08 | 55.84 |
| +entity type | 88.89 | 46.92 | 65.12 | 76.82 | 71.49 |
| +token idf | **89.79** | **48.80** | **65.69** | **77.15** | **73.24** |

Table 2: Ablation study of the proposed method using LLaMA-30b ($\gamma = 0.9, \rho = 0.01$).

tricks serves as the baseline in the first row, with each trick incorporated incrementally in the succeeding rows. The ablation studies on the remaining 21 proxy models are detailed in Appendix H.

**Focus on the informative keywords**. By focusing on the keywords, improvements are observed across nearly all metrics. Notably, the Pearson and Spearman correlation coefficients are improved by 5.04% and 7.48%, respectively. These results suggest a stronger correlation between the keyword uncertainties and passage-level human judgments.

**Focus on the preceding words**. When hallucination propagation is incorporated on the basis of keyword selection, remarkably, substantial improvements can be observed across all metrics. Particularly, the AUC-PR of the non-factual class exhibited a significant increase of 3.67% on LLaMA-30b. This enhancement can be attributed to the successful remediation of the overconfidence issue as discussed in Section 3.2.

The overall performance of LLaMA-30b with $\gamma$ ranging from 0 to 1 (no hallucination propagation when $\gamma$ is set to zero) is illustrated in Figure 10 of

| Text | $h$ | $\hat{h}$ |
|------|-----|-----------|
| Paul Taylor is an American singer-songwriter, multi-instrumentalist, and record producer. He is best known as the lead singer and songwriter of the band Winger. | 12.38 | 133.96 |
| C. V. Ananda Bose was an Indian freedom fighter, lawyer, and politician. (...) He was a member of the Indian delegation to the United Nations in 1951. He was a member of the Indian delegation to the United Nations in 1952. | 1.36 | 53.53 |

Table 3: Cases detected by hallucination propagation. $h$ and $\hat{h}$ denote the hallucination scores of the highlighted sentences without and with penalty, respectively.

Appendix G. It is evident that most metrics improve as $\gamma$ increases. However, a performance decline is noticeable when $\gamma$ exceeds 0.8, indicating that an excessive focus on the preceding words could also lead to a deterioration in performance.

**Focus on the token properties**. Further enhancements in model performance can be achieved by incorporating entity type information and token IDF, leading to drastic improvements as evidenced in the last two rows. Specifically, the AUC-PR of the factual class increases by 10.76%, and both correlation coefficients improve by approximately 18%. This demonstrates the effectiveness of probability correction in mitigating the underconfidence problem as discussed in Section 3.3. Nevertheless, we observe little improvement for the non-factual* class when considering only the entity type property on multiple proxy models. The reason behind this observation will be explained in Section 4.4.2.

The performance impact of varying $\rho$ values is depicted in Figure 11 of Appendix G. Generally, $\rho = 0.01$ delivers optimal results. A large $\rho$ could lead to the omission of crucial information due to a restricted candidate set, while a small $\rho$ might introduce noise by including irrelevant tokens.

## 4.4 Case study

### 4.4.1 Non-factual cases detected by hallucination propagation

Table 3 showcases two examples of hallucinated content accurately identified by hallucination propagation. In the first case, the pink sentence erroneously assigns the role of singer and songwriter to Paul Taylor, who was actually a keyboardist/guitarist of the band Winger. This error originates from the model's preceding hallucination (purple text)

| | $\overline{h}$ without type | $\overline{h}$ with type |
|---|---|---|
| major-inaccurate | 14.99 | **4.09** |
| minor-inaccurate | 9.70 | **3.79** |
| accurate* | 5.63 | 2.75 |

Table 4: The average hallucination scores for each category with and without entity type information provided.

"Paul Taylor is an American singer-songwriter". In the second case, the pink sentence duplicates existing text, consequently producing a significantly low value of $h$ owing to the overconfidence problem. With the introduction of the penalty, the hallucination score increases by approximately fifty-fold, demonstrating the effectiveness of focusing on the hallucination scores of the preceding words.

The attention heat maps corresponding to the two cases can be found in Appendix C.

### 4.4.2 Failure cases after entity type provision

To explain the decrease in AUC-PR of the non-factual* class when entity types are specified for each named entity, we computed the sentence-level average hallucination score $\overline{h}$ for each category in Table 4.

We notice that the average hallucination score $\overline{h}$ for all classes decreases when entity type information is provided, since the probability is corrected to be more confident for the keywords. However, this decrease is especially noticeable in the major inaccurate category due to the fact that sentences labeled as major inaccurate contain more hallucinated keywords. As a result, distinguishing between major inaccurate and minor inaccurate becomes more challenging. Given that the non-factual* class only includes sentences classified as major inaccurate, this increased difficulty in differentiation contributes to the observed decrease in AUC-PR for the non-factual* class.

## 5 Conclusion

In this paper, we propose a reference-free, uncertainty-based method for detecting hallucinations in LLMs. The proposed method aims to imitate human factuality checking by considering three aspects: focus on informative keywords, focus on preceding words and focus on token properties. Our experimental results empirically demonstrate the effectiveness of the proposed method for hallucination detection at both sentence and passage level, without requiring any external knowledge or training data. We have also analyzed how each of

the three focus mechanisms impacts the overall performance when using different proxy models as the backbone. The results on XSumFaith and FRANK datasets further showcase the potential capability of the proposed method for detecting hallucinations produced by small models. We hope our work can contribute to the field of LLM research and help improve the reliability and factuality of LLMs.

## Limitations

The keyword identification and named entity recognition in our approach is based on Spacy, which may introduce some errors as observed in our practice. For instance, the television drama "The Great Ambition" could erroneously be classified as an organization. Such failures can result in the calculated probability becoming unreliable, leading to a decrease in performance. Additionally, the categories of named entities in real-world scenarios are considerably more diverse than those identifiable by Spacy, such as food, vehicles, and other specialized domains.

A further limitation arises from our assumption that LLM proxies are consistently current with factual knowledge. However, LLMs are not continuously updated post-training, hence they may lack recently emerged factual information. This could influence the assigned probabilities and in turn affect our hallucination detection's effectiveness.

## Ethics Statement

We maintained privacy in our approach, as our method does not require user data and we conducted experiments on publicly available datasets, upholding privacy and anonymity standards. Despite the intention of this work is to improve LLMs' reliability, potential misuse such as using it to enhance AI-generated misinformation or deepfake content, is condemned. We are dedicated to ethical AI research, addressing potential concerns, and maintaining a balance between technological progress and ethical responsibility.

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

## A  Results for detecting hallucinations generated by small models

To assess the effectiveness of our proposed method in detecting hallucinations produced by small models, we conducted experiments using two hallucination detection datasets extracted from the test split of the SummaC benchmark (Laban et al., 2022): XSumFaith and FRANK. These datasets consist of summaries generated by small models such as TransS2S (Vaswani et al., 2017), TCONVS2S (Narayan et al., 2018), and BART (Lewis et al., 2020).

Although the benchmark includes a total of six datasets, it should be noted that some of them contain summarizations not produced by generative models, such as Polytope (Huang et al., 2020) and SummEval (Fabbri et al., 2021). Additionally, certain datasets (Kryściński et al., 2020; Falke et al., 2019) label any content not present in the input as extrinsic hallucination[5]. However, such extrinsic hallucination might actually be factual (Dong et al., 2022; Cao et al., 2022).

---

[5]Only about 5% of such cases in XSumFaith as reported in the original paper, hence, we have included this dataset in our analysis.

Specifically, for the FRANK dataset, which provides the error type of each sample, we removed the instances that were labeled as OutE (statement contains information not present in the source article) for the reason discussed above. For XSumFaith, we excluded the human-written summaries since they may differ in style from model-generated summaries (Gekhman et al., 2023). The statistics of the two datasets are shown in Table 5.

| Dataset | #Num | %Hallucination |
|---------|------|----------------|
| XSumFaith | 984 | 90.40 |
| FRANK | 1242 | 57.41 |

Table 5: The statistics of XSumFaith and FRANK dataset (test split from SummaC benchmark).

We report the AUC-PR for the non-factual class and factual class and balanced-accuracy in Table 6 and Table 7. Our method performs well across all three metrics when applied to the XSumFaith dataset. However, we observed that, for the FRANK dataset, using only the negative log probability yields better results compared to using the sum of negative log probability and entropy. Furthermore, focus on the keywords proves less effective than considering all tokens in the passage. We attribute this discrepancy to the unique characteristics of the FRANK dataset, which contains hallucinations such as predicate errors, pronoun errors, and preposition errors. Therefore, we only use the negative log probability of token $t$ as its hallucination score and disregard keyword selection for FRANK dataset. These results highlight the effectiveness of focusing on token property information, but little enhancement is observed when solely relying on hallucination propagation. Further investigation is left for the future work.

| Method | NonFact | Fact | Balanced-Acc |
|--------|---------|------|--------------|
| avg($h$) | 92.79 | 11.75 | 57.65 |
| +keyword | 92.65 | 14.19 | 56.24 |
| +penalty | 92.34 | 14.97 | 57.77 |
| +entity type | 94.98 | 18.46 | 64.77 |
| +token idf | **95.13** | **18.86** | **64.81** |

Table 6: Performance of the proposed method using LLaMA-30b-SFT on XSumFaith dataset ($\gamma = 0.9, \rho = 0.01$).

| Method | NonFact | Fact | Balanced-Acc |
|---|---|---|---|
| avg($-logp$) | 89.82 | 79.00 | 78.79 |
| +penalty | 89.87 | 78.37 | 79.46 |
| +entity type | **90.44** | 79.78 | 80.31 |
| +token idf | 90.12 | **80.00** | **80.70** |

Table 7: Performance of the proposed method using LLaMA-30b-SFT on FRANK dataset ($\gamma = 0.4, \rho = 0.01$).

## B Implementation Details

Our experiments were conducted on an AWS p3dn.24xlarge instance, each of which is equipped with 8 NVIDIA V100 32GiB GPUs, 96 CPU cores, and 768 GiB RAM. In order to prevent the influence of type tags when calculating the token probability, we set the probability of token "<" to zero. When using the SFT version of LLaMA, the prompt as described in Section 4.1 is formatted to follow the Alpaca (Taori et al., 2023) pattern: "### Instruction: {instruction} ### Response: {response}".

For the experiments on the two summarization datasets, we excluded instances where the token count exceeded LLaMA's maximum context length of 2048, resulting in the elimination of 16 cases from the XSumFaith dataset. The prompts employed are "{document} TL;DR" and "Summarize the following text using appropriate words that follow to the given type: {document} TL;DR", without and with the provision of entity types, respectively.

Entity types are also provided in the prompts as few-shot examples. For instance, the prompt for the concept "michael savage" is "This a passage from <ORG> Wikipedia about <PERSON> michael savage:".

## C More attention heat map cases

Figure 4 and Figure 5 provide visualizations of the attention heat maps of the two cases mentioned in Section 4.4.1. The attentions that are erroneously directed towards preceding unreliable tokens are marked within a blue box.

## D Examples of passages with entity types provided

Figure 6 to Figure 8 illustrate three examples of Wikipedia passages generated by text-davinci-003, along with their corresponding prompts. Before

inputting each passage into the proxy model for hallucination detection, the entity types are provided before each named entity recognized by Spacy.

## E Details of the proxy models

The 22 proxy models used in our experiments include LLaMA-{7b, 13b, 30b, 65b} (Touvron et al., 2023a), LLaMA-2{7b, 13b, 70b} (Touvron et al., 2023b), OPT-{125m, 1.3b, 13b, 30b} (Zhang et al., 2022), GPT-J-6b (Wang and Komatsuzaki, 2021) GPT-NeoX-20b (Black et al., 2022), Falcon-{7b, 40b} (Almazrouei et al., 2023), Vicuna-{7b, 13b, 33b} (Chiang et al., 2023), RedPajama-{3b, 7b} (Computer, 2023) and instruction tuning versions of LLaMA-{13b, 30b}-SFT[6].

## F Performance comparison of LLaMA family

Figure 9 presents the performance comparison among the LLaMA family. Models with a larger parameter size generally demonstrate superior performance on the WikiBio GPT-3 dataset. However, despite being twice the size of LLaMA-30b, LLaMA-65b underperforms across four out of the five evaluated metrics compared to LLaMA-30b.

## G Hyper parameters analysis

Figure 10 shows the performance of LLaMA-30b with $\gamma$ ranging from 0 to 1. When $\gamma$ is set to zero, no penalty is accumulated to the token hallucination score. Figure 11 depicts the performance impact of varying $\rho$. Setting $\rho$ either too large or too small leads to a decrease in performance.

## H Additional Results

The main results including all the 22 proxy models are shown in Table 8. As observed in Table 9 to Table 29, our method consistently outperforms the performance achieved by solely relying on the uncertainty metric. The optimal setting may vary across models, we attribute this to the different generation patterns exhibited by each model.

---

[6]https://huggingface.co/ausboss/llama-30b-supercot

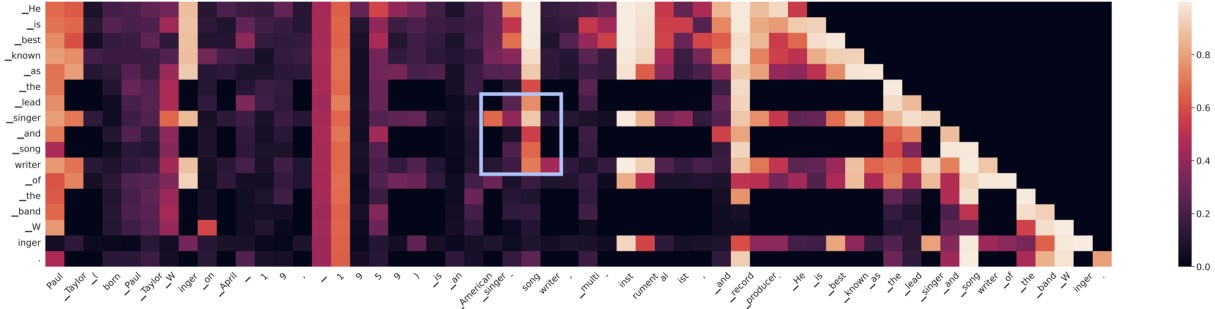

Figure 4: The attention heat map corresponding to the first case in Section 4.4.1. Due to space limitations, not all sentences are depicted in the figure.

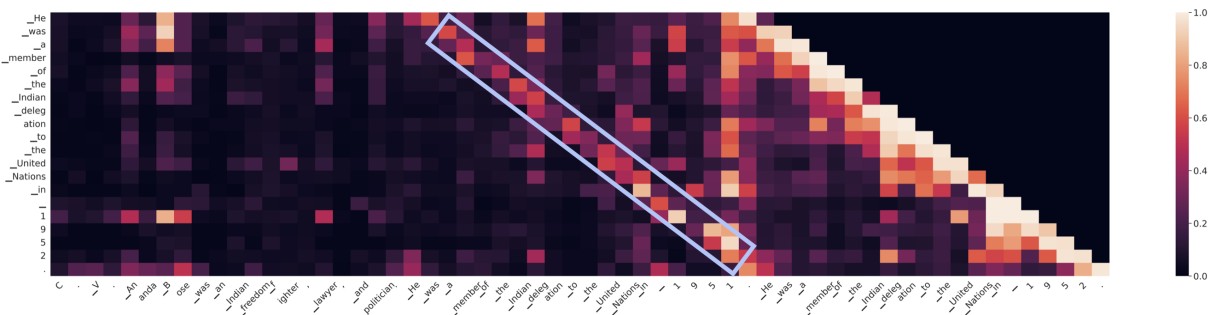

Figure 5: The attention heat map corresponding to the second case in Section 4.4.1. Due to space limitations, not all sentences are depicted in the figure.

**Case 1: Michael Savage**

```
Please complete the passage below using appropriate words that follow to the given type with < >
↪    wrapped.

This is a passage from <ORG> Wikipedia about <PERSON> michael savage:
<PERSON> Michael Alan Weiner (born <DATE> March 31, 1942), better known by his professional name
↪    <PERSON> Michael Savage, is an <NORP> American radio host, author, activist, nutritionist, and
↪    conservative political commentator. He is the host of <ORG> The Savage Nation, a nationally
↪    syndicated talk show that aired on <ORG> Talk Radio Network across <GPE> the United States
↪    until <DATE> 2012, and in <DATE> 2009 was the <ORDINAL> second most listened-to radio talk
↪    show in the country with an audience of <CARDINAL> over 20 million listeners on <CARDINAL> 400
↪    stations across <GPE> the United States. Since <DATE> October 23, 2012, <PERSON> Michael
↪    Savage has been syndicated by <ORG> Cumulus Media Networks. He holds master's degrees from
↪    <ORG> the University of Hawaii in medical botany and medical anthropology, and a <WORK_OF_ART>
↪    Ph.D. from <ORG> the University of California, Berkeley in nutritional ethnomedicine. As
↪    <PERSON> Michael Weiner, he has written books on nutrition, herbal medicine, and homeopathy.
```

Figure 6: The text-davinci-003 generated Wikipedia passage about Michael Savage in WikiBio GPT-3 dataset.

Figure 7: The text-davinci-003 generated Wikipedia passage about Michael Replogle in WikiBio GPT-3 dataset.

Figure 8: The text-davinci-003 generated Wikipedia passage about Tommy Nutter in WikiBio GPT-3 dataset.

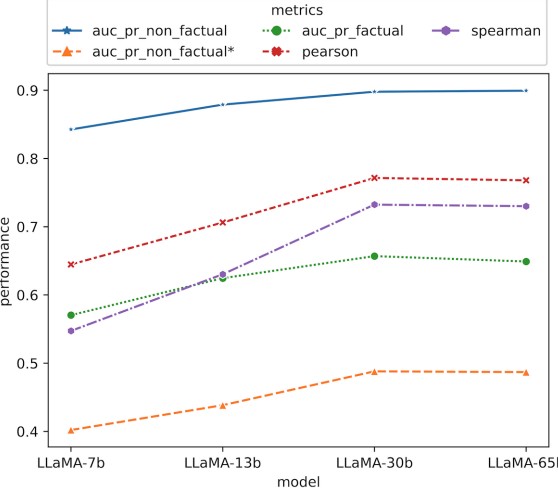

Figure 9: Performance comparison of LLaMA family with varying parameter sizes.

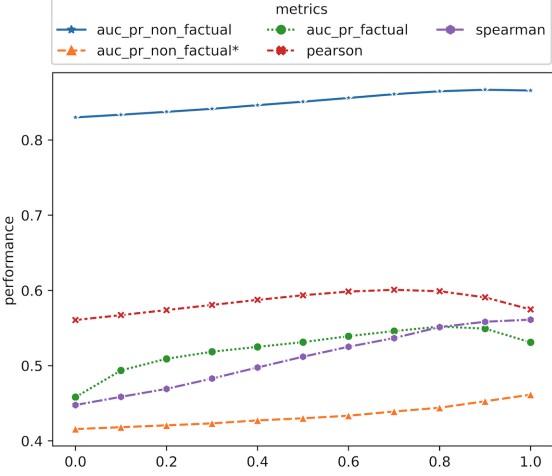

Figure 10: Performance of LLaMA-30b with different $\gamma$.

| Method | Sentence-level Metrics | | | Passage-level Metrics | |
| | NonFact | NonFact* | Factual | Pearson | Spearman |
|---|---|---|---|---|---|
| **GPT-3 Uncertainties** | | | | | |
| Avg($-\log p$) | 83.21 | 38.89 | 53.97 | 57.04 | 53.93 |
| Avg($\mathcal{H}$) | 80.73 | 37.09 | 52.07 | 55.52 | 50.87 |
| Max($-\log p$) | 87.51 | 35.88 | 50.46 | 57.83 | 55.69 |
| Max($\mathcal{H}$) | 85.75 | 32.43 | 50.27 | 52.48 | 49.55 |
| **SelfCheckGPT** | | | | | |
| BERTScore | 81.96 | 45.96 | 44.23 | 58.18 | 55.90 |
| QA | 84.26 | 40.06 | 48.14 | 61.07 | 59.29 |
| Unigram (max) | 85.63 | 41.04 | 58.47 | 64.71 | 64.91 |
| Combination | 87.33 | 44.37 | 61.83 | 69.05 | 67.77 |
| **Ours** | | | | | |
| GPT-J-6B$_{focus}$ | 77.92 | 38.56 | 33.58 | 15.68 | 13.95 |
| GPT-NeoX-20B$_{focus}$ | 81.40 | 38.84 | 39.80 | 35.03 | 30.40 |
| OPT-125M$_{focus}$ | 73.88 | 34.74 | 28.29 | -8.04 | -5.92 |
| OPT-1.3B$_{focus}$ | 73.84 | 34.00 | 30.88 | 1.08 | -1.20 |
| OPT-13B$_{focus}$ | 79.63 | 39.97 | 39.23 | 27.88 | 23.65 |
| OPT-30B$_{focus}$ | 79.26 | 39.49 | 40.63 | 31.07 | 28.67 |
| Falcon-7B$_{focus}$ | 82.25 | 40.94 | 41.25 | 45.19 | 36.76 |
| Falcon-40B$_{focus}$ | 88.11 | 46.95 | 58.14 | 68.63 | 64.66 |
| Vicuna-7b$_{focus}$ | 84.14 | 39.93 | 53.41 | 58.78 | 49.84 |
| Vicuna-13b$_{focus}$ | 86.87 | 41.80 | 60.25 | 66.72 | 58.64 |
| Vicuna-33b$_{focus}$ | 88.23 | 44.51 | 62.10 | 71.82 | 65.96 |
| RedPajama-3B$_{focus}$ | 82.26 | 40.49 | 43.38 | 46.48 | 40.56 |
| RedPajama-7B$_{focus}$ | 84.68 | 41.53 | 50.05 | 55.55 | 49.74 |
| LLaMA-7B$_{focus}$ | 84.26 | 40.20 | 57.04 | 64.47 | 54.73 |
| LLaMA-2-7B$_{focus}$ | 84.29 | 41.31 | 56.64 | 63.59 | 48.91 |
| LLaMA-13B$_{focus}$ | 87.90 | 43.84 | 62.46 | 70.62 | 63.03 |
| LLaMA-2-13B$_{focus}$ | 87.28 | 45.62 | 63.39 | 71.57 | 63.85 |
| LLaMA-13B-SFT$_{focus}$ | 88.17 | 44.62 | 62.25 | 71.91 | 63.81 |
| LLaMA-30B$_{focus}$ | 89.79 | 48.80 | **65.69** | 77.15 | **73.24** |
| LLaMA-30B-SFT$_{focus}$ | **90.34** | 49.17 | 65.29 | **77.53** | 73.10 |
| LLaMA-65B$_{focus}$ | 89.94 | 48.69 | 64.90 | 76.80 | 73.01 |
| LLaMA-2-70B$_{focus}$ | 89.95 | **52.06** | 65.11 | 76.88 | 72.36 |

Table 8: Main results including all proxy models in Section 4.1.

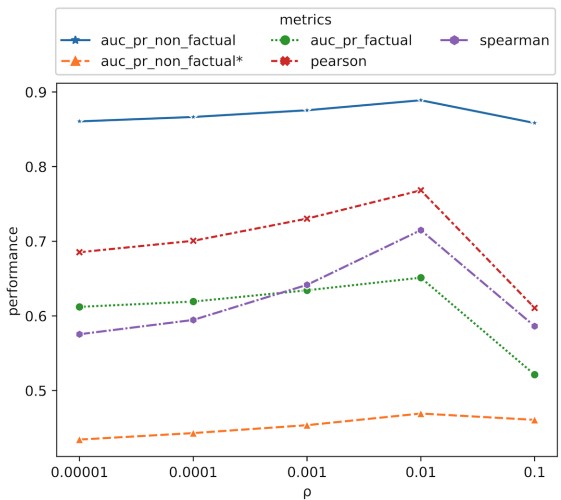

Figure 11: Performance of LLaMA-30b with different $\rho$.

| Method | NoFac | NoFac* | Fact | Pear. | Spear. |
|---|---|---|---|---|---|
| avg($h$) | 78.67 | 37.19 | 35.81 | 29.11 | 16.65 |
| +keyword | 79.27 | 37.28 | 35.86 | 36.26 | 23.61 |
| +penalty | 83.48 | **43.77** | 49.22 | 46.22 | 38.32 |
| +entity type | 83.40 | 39.27 | 56.36 | 62.57 | 52.63 |
| +token idf | **84.26** | 40.20 | **57.04** | **64.47** | **54.73** |

Table 9: Ablation study of the proposed method using LLaMA-7b ($\gamma = 0.9, \rho = 0.01$).

| Method | NoFac | NoFac* | Fact | Pear. | Spear. |
|---|---|---|---|---|---|
| avg($h$) | 80.18 | 38.57 | 41.21 | 39.80 | 25.99 |
| +keyword | 80.93 | 38.85 | 40.31 | 45.83 | 32.37 |
| +penalty | 83.26 | 43.02 | 43.35 | 37.92 | 33.99 |
| +entity type | 87.12 | 43.13 | 61.72 | 69.99 | 60.64 |
| +token idf | **87.90** | **43.84** | **62.46** | **70.62** | **63.03** |

Table 10: Ablation study of the proposed method using LLaMA-13b ($\gamma = 0.9, \rho = 0.01$).

| Method | NoFac | NoFac* | Fact | Pear. | Spear. |
|---|---|---|---|---|---|
| avg($h$) | 82.62 | 40.94 | 48.74 | 52.60 | 40.85 |
| +keyword | 83.64 | 41.00 | 46.77 | 58.01 | 49.44 |
| +penalty | 88.06 | 46.94 | 49.49 | 54.92 | 56.69 |
| +entity type | 89.54 | 47.66 | 64.27 | 76.30 | 72.54 |
| +token idf | **89.94** | **48.69** | **64.90** | **76.80** | **73.01** |

Table 11: Ablation study of the proposed method using LLaMA-65b ($\gamma = 0.9, \rho = 0.01$).

| Method | NoFac | NoFac* | Fact | Pear. | Spear. |
|---|---|---|---|---|---|
| avg($h$) | 80.00 | 39.28 | 41.22 | 39.36 | 24.80 |
| +keyword | 81.01 | 39.29 | 41.14 | 46.43 | 31.94 |
| +penalty | 84.39 | **45.06** | 51.36 | 48.81 | 40.78 |
| +entity type | 87.81 | 44.28 | 61.81 | 71.54 | 62.75 |
| +token idf | **88.17** | 44.62 | **62.25** | **71.91** | **63.81** |

Table 12: Ablation study of the proposed method using LLaMA-13b-SFT ($\gamma = 0.9, \rho = 0.01$).

| Method | NoFac | NoFac* | Fact | Pear. | Spear. |
|---|---|---|---|---|---|
| avg($h$) | 81.58 | 42.19 | 47.56 | 49.02 | 35.50 |
| +keyword | 83.32 | 42.63 | 47.13 | 56.45 | 45.38 |
| +penalty | 86.95 | 45.74 | 59.60 | 66.56 | 59.14 |
| +entity type | 89.92 | 48.52 | 65.12 | 77.48 | 72.42 |
| +token idf | **90.34** | **49.17** | **65.29** | **77.53** | **73.10** |

Table 13: Ablation study of the proposed method using LLaMA-30b-SFT ($\gamma = 0.9, \rho = 0.01$).

| Method | NoFac | NoFac* | Fact | Pear. | Spear. |
|---|---|---|---|---|---|
| avg($h$) | 77.06 | 35.47 | 29.42 | 9.64 | 3.88 |
| +keyword | 77.59 | 35.61 | 30.41 | 19.24 | 10.98 |
| +penalty | 80.74 | **42.23** | 38.03 | 24.18 | 20.17 |
| +entity type | 81.66 | 41.55 | 40.03 | 44.04 | 36.62 |
| +token idf | **82.25** | 40.94 | **41.25** | **45.19** | **36.76** |

Table 14: Ablation study of the proposed method using Falcon-7b ($\gamma = 1.0, \rho = 0.01$).

| Method | NoFac | NoFac* | Fact | Pear. | Spear. |
|---|---|---|---|---|---|
| avg($h$) | 79.72 | 37.50 | 32.37 | 34.00 | 27.47 |
| +keyword | 80.55 | 37.62 | 35.13 | 45.45 | 38.11 |
| +penalty | 87.26 | 47.22 | 44.88 | 47.67 | 52.01 |
| +entity type | 87.11 | 45.74 | 57.60 | 68.25 | 62.46 |
| +token idf | **88.11** | **46.95** | **58.14** | **68.63** | **64.66** |

Table 15: Ablation study of the proposed method using Falcon-40b ($\gamma = 0.9, \rho = 0.01$).

| Method | NoFac | NoFac* | Fact | Pear. | Spear. |
|---|---|---|---|---|---|
| avg($h$) | 76.29 | 27.08 | 29.65 | 11.28 | 3.51 |
| +keyword | 77.50 | 27.56 | 32.85 | 19.50 | 8.34 |
| +penalty | 74.59 | 33.83 | 38.53 | 23.30 | 9.87 |
| +entity type | 83.14 | 38.45 | 52.50 | 56.32 | 48.41 |
| +token idf | **84.14** | **39.93** | **53.41** | **58.78** | **49.84** |

Table 16: Ablation study of the proposed method using Vicuna-7b ($\gamma = 0.9, \rho = 0.01$).

| Method | NoFac | NoFac* | Fact | Pear. | Spear. |
|---|---|---|---|---|---|
| avg($h$) | 79.21 | 36.42 | 35.44 | 27.53 | 17.35 |
| +keyword | 80.58 | 36.46 | 37.10 | 40.37 | 27.67 |
| +penalty | 84.52 | **43.10** | 56.41 | 51.52 | 40.13 |
| +entity type | 86.78 | 41.35 | 59.96 | **67.24** | **58.90** |
| +token idf | **86.87** | 41.80 | **60.25** | 66.72 | 58.64 |

Table 17: Ablation study of the proposed method using Vicuna-13b ($\gamma = 0.9, \rho = 0.01$).

| Method | NoFac | NoFac* | Fact | Pear. | Spear. |
|---|---|---|---|---|---|
| avg($h$) | 81.96 | 41.91 | 42.83 | 42.09 | 31.30 |
| +keyword | 82.95 | 40.87 | 42.90 | 49.91 | 39.35 |
| +penalty | 86.59 | **47.65** | 61.02 | 62.39 | 55.71 |
| +entity type | 88.10 | 44.40 | 61.35 | 71.06 | 64.53 |
| +token idf | **88.23** | 44.51 | **62.10** | **71.82** | **65.96** |

Table 18: Ablation study of the proposed method using Vicuna-33b ($\gamma = 0.9, \rho = 0.01$).

| Method | NoFac | NoFac* | Fact | Pear. | Spear. |
|---|---|---|---|---|---|
| avg($h$) | 77.48 | 32.96 | 30.24 | 18.92 | 5.91 |
| +keyword | 78.32 | 34.00 | 32.15 | 28.78 | 14.05 |
| +penalty | **82.36** | 42.41 | **47.25** | 42.65 | 26.51 |
| +entity type | 82.02 | 40.46 | 43.24 | 46.31 | 39.44 |
| +token idf | 82.26 | 40.49 | 43.38 | **46.48** | **40.56** |

Table 19: Ablation study of the proposed method using RedPajama-3b ($\gamma = 1.0, \rho = 0.01$).

| Method | NoFac | NoFac* | Fact | Pear. | Spear. |
|---|---|---|---|---|---|
| avg($h$) | 79.43 | 34.37 | 33.22 | 36.82 | 22.56 |
| +keyword | 80.33 | 35.46 | 35.92 | 44.25 | 30.43 |
| +penalty | 83.57 | 41.33 | 44.87 | 42.34 | 35.39 |
| +entity type | 84.34 | 40.87 | **50.53** | **56.38** | **50.28** |
| +token idf | **84.68** | **41.53** | 50.05 | 55.55 | 49.74 |

Table 20: Ablation study of the proposed method using RedPajama-7b ($\gamma = 0.9, \rho = 0.01$).

| Method | NoFac | NoFac* | Fact | Pear. | Spear. |
|---|---|---|---|---|---|
| avg($h$) | 75.64 | 33.34 | 28.30 | -0.38 | -9.30 |
| +keyword | 76.31 | 33.99 | 29.61 | 9.26 | -2.55 |
| +penalty | 77.51 | 38.05 | **37.54** | **25.50** | 7.06 |
| +entity type | 77.68 | 37.73 | 33.98 | 17.82 | **15.29** |
| +token idf | **77.92** | **38.56** | 33.58 | 15.68 | 13.95 |

Table 21: Ablation study of the proposed method using GPT-J-6b ($\gamma = 1.0, \rho = 0.01$).

| Method | NoFac | NoFac* | Fact | Pear. | Spear. |
|---|---|---|---|---|---|
| avg($h$) | 77.14 | 33.49 | 30.71 | 11.55 | 3.18 |
| +keyword | 77.97 | 33.70 | 32.93 | 23.84 | 11.53 |
| +penalty | 80.77 | **40.22** | **43.01** | **37.46** | 23.13 |
| +entity type | 80.12 | 37.50 | 38.70 | 31.24 | 25.08 |
| +token idf | **81.40** | 38.84 | 39.80 | 35.03 | **30.40** |

Table 22: Ablation study of the proposed method using GPT-NeoX-20b ($\gamma = 1.0, \rho = 0.01$).

| Method | NoFac | NoFac* | Fact | Pear. | Spear. |
|---|---|---|---|---|---|
| avg($h$) | 71.05 | 30.96 | 24.64 | -19.84 | -23.08 |
| +keyword | 71.81 | 32.65 | 25.08 | -15.16 | -19.79 |
| +penalty | 72.71 | **37.25** | 26.19 | -9.94 | -14.61 |
| +entity type | 73.64 | 34.64 | 28.12 | -9.09 | -6.75 |
| +token idf | **73.88** | 34.74 | **28.29** | **-8.04** | **-5.92** |

Table 23: Ablation study of the proposed method using OPT-125m ($\gamma = 1.0, \rho = 0.01$).

| Method | NoFac | NoFac* | Fact | Pear. | Spear. |
|---|---|---|---|---|---|
| avg($h$) | 73.73 | 32.15 | 26.00 | -11.16 | -17.54 |
| +keyword | 74.32 | 33.51 | 27.05 | -3.51 | -13.54 |
| +penalty | **74.84** | **37.37** | **31.13** | **4.08** | -8.62 |
| +entity type | 73.50 | 33.77 | 30.02 | -1.03 | -2.59 |
| +token idf | 73.84 | 34.00 | 30.88 | 1.08 | **-1.20** |

Table 24: Ablation study of the proposed method using OPT-1.3b ($\gamma = 1.0, \rho = 0.01$).

| Method | NoFac | NoFac* | Fact | Pear. | Spear. |
|---|---|---|---|---|---|
| avg($h$) | 76.77 | 33.82 | 29.75 | 4.36 | -3.96 |
| +keyword | 77.40 | 34.44 | 31.67 | 13.79 | 1.89 |
| +penalty | **79.72** | 39.44 | **40.65** | **29.13** | 14.30 |
| +entity type | 79.06 | 39.19 | 38.74 | 28.42 | 22.59 |
| +token idf | 79.63 | **39.97** | 39.23 | 27.88 | **23.65** |

Table 25: Ablation study of the proposed method using OPT-13b ($\gamma = 1.0, \rho = 0.01$).

| Method | NoFac | NoFac* | Fact | Pear. | Spear. |
|---|---|---|---|---|---|
| avg($h$) | 76.98 | 33.77 | 29.65 | 5.52 | -1.62 |
| +keyword | 77.61 | 34.59 | 31.55 | 14.25 | 2.98 |
| +penalty | **79.67** | **39.91** | **43.18** | 31.42 | 16.32 |
| +entity type | 79.31 | 39.32 | 41.66 | **33.75** | **28.98** |
| +token idf | 79.26 | 39.49 | 40.63 | 31.07 | 28.67 |

Table 26: Ablation study of the proposed method using OPT-30b ($\gamma = 1.0, \rho = 0.01$).

| Method | NoFac | NoFac* | Fact | Pear. | Spear. |
|---|---|---|---|---|---|
| avg($h$) | 77.04 | 34.22 | 30.08 | 28.83 | 13.62 |
| +keyword | 78.20 | 35.64 | 33.23 | 38.39 | 23.96 |
| +penalty | 84.02 | **42.90** | 39.17 | 39.10 | 41.84 |
| +entity type | 83.55 | 41.12 | 55.57 | 62.05 | 47.94 |
| +token idf | **84.29** | 41.31 | **56.64** | **63.59** | **48.91** |

Table 27: Ablation study of the proposed method using LLaMA-2-7b ($\gamma = 0.9, \rho = 0.01$).

| Method | NoFac | NoFac* | Fact | Pear. | Spear. |
|---|---|---|---|---|---|
| avg($h$) | 77.71 | 34.31 | 32.24 | 36.55 | 22.61 |
| +keyword | 79.75 | 36.24 | 35.22 | 47.11 | 34.73 |
| +penalty | 84.36 | 43.64 | 51.50 | 52.88 | 44.72 |
| +entity type | 85.87 | 43.46 | 63.20 | 71.24 | 59.62 |
| +token idf | **87.28** | **45.62** | **63.39** | **71.57** | **63.85** |

Table 28: Ablation study of the proposed method using LLaMA-2-13b ($\gamma = 0.9, \rho = 0.01$).

| Method | NoFac | NoFac* | Fact | Pear. | Spear. |
|---|---|---|---|---|---|
| avg($h$) | 79.05 | 37.35 | 36.99 | 49.79 | 39.58 |
| +keyword | 81.50 | 39.17 | 40.78 | 59.18 | 51.97 |
| +penalty | 86.76 | 46.39 | 50.34 | 55.53 | 58.53 |
| +entity type | 89.66 | 51.33 | **65.14** | **77.58** | **72.43** |
| +token idf | **89.95** | **52.06** | 65.11 | 76.88 | 72.36 |

Table 29: Ablation study of the proposed method using LLaMA-2-70b ($\gamma = 0.9, \rho = 0.01$).