# OpenReview forum: "Enhancing Uncertainty-Based Hallucination Detection with Stronger Focus"
_EMNLP/2023/Conference — EMNLP 2023 Main_

### Official Review · Reviewer_2xZc · 2023-08-05

**Soundness:** 4

**Excitement:**

4: Strong: This paper deepens the understanding of some phenomenon or lowers the barriers to an existing research direction.

**Justification For Ethical Concerns:**

N.A.

**Missing References:**

N.A.

**Paper Topic And Main Contributions:**

Hallucination plagues the LLM progress and makes them inviable for industry use. Hallucination detection is either based on retrieval of knowledge base or expensive sampling of answers from the LLM. Both these methods have limitations. The paper aims to address these limitations by proposing a novel reference free uncertainty based method for detecting hallucinations. They demonstrate through experiments, human evaluations and ablative analysis that their approach is effective and beats performances by other systems.

**Questions For The Authors:**

N.A.

**Reasons To Accept:**

1: Explains the class of hallucination methods, why they are not perfect.
2: Identifies aspects of human behavior used in resolving hallucination.
3: Explains the concepts through well reasoned text, equations and diagrams.
4: Performs good experiments, satisfying ablation studies and achieve results higher than baselines. Also appropriate baselines selected.
5: Fair on limitations - I was impressed.

**Reasons To Reject:**

N.A.

**Reproducibility:**

4: Could mostly reproduce the results, but there may be some variation because of sample variance or minor variations in their interpretation of the protocol or method.

**Reviewer Confidence:**

4: Quite sure. I tried to check the important points carefully. It's unlikely, though conceivable, that I missed something that should affect my ratings.

**Typos Grammar Style And Presentation Improvements:**

Please explain properly what focus means? The use of phrases like line 291 "...we focus on hallucination scores of preceding tokens by..." or line 110 "... to strengthen the focus on hallucination".  use focus in an unnatural way. To me it feels like the use of the word "attention" in transformers paper. But in the latter, they gave a good definition of it. This, otherwise well written paper, was difficult to comprehend every time focus was mentioned unnaturally.

Please explain exposure bias in lines 265-270 by giving example. People may not be aware of it and it is a key point that must be elaborated.

Please rephrase lines 272 to 287. They make it seem like the figure 2 will show probabilities 0.49 and 0.90 but it confuses the reader when the figure shows no such thing. I get the points you are trying to make here but it should be rephrased.

---

> ### Author Rebuttal · Authors · 2023-08-29
>
> Thank you for your insightful review. We'd like to address your valuable feedback below:
>
> + Indeed, "focus" is a pivotal concept in our work, we intended to convey the idea of "focus" as "highlighting" or "prioritizing" certain aspects within the context of hallucination detection, similar to how a human might emphasize certain aspects when detecting hallucination. We agree that the term "focus" might have been employed in an unnatural manner, leading to confusion. To alleviate this, we will rephrase the sections you pointed out in the final version, ensuring that the meaning and purpose of "focus" are explicitly and intuitively explained.
> + We agree with your comments about exposure bias (lines 265\~270). In the final version, we will provide an example of this concept and its significance in our work to enhance the clarity of our presentation.
> + We acknowledge that this is an important point, and we will represent both the 2 probabilities, 0.49 and 0.9, in **Figure 1** in the final version, and carefully revise the wording of lines 272\~287 to ensure that the conveyed message is aligned with the information presented in the figure.
>
> Thank you again for your suggestions and we will ensure that our final version will effectively address the concerns you raised.

---

### Official Review · Reviewer_3BQy · 2023-08-05

**Soundness:** 4

**Excitement:**

4: Strong: This paper deepens the understanding of some phenomenon or lowers the barriers to an existing research direction.

**Paper Topic And Main Contributions:**

This paper focuses on model based hallucination detection approach. Different from sampling based or retrieval based approach, this paper adopts uncertainty based approach to calculate the hallucination score. Big models are used as proxy language model to calculate the generation probability of each term, which lays the foundation to calculate the uncertainty score. However, a simple setup of the proxy model does not suffice to succeed, which either present "overconfident" or "underconfident" behavior.

From experiment observation, authors of this paper find that (1) there is strong attention relationship among the hallucinated terms, which leads to the "overconfident" behavior to generate more hallucination content; (2) while there is difference in next word prediction between real human's mindset and model-based approach. Real person is prone to generate the next word based on semantic info of the context such as entity type to narrow down the next word choice into a small space, but the model would consider and sum up all possibilities to predict next word, which makes the correct next word under-scored. (underconfident).

Based on these observation, authors in this paper propose (1) penalty on the strongly connected hallucinated words; (2) adjust the under scored term based on term statistic properties. To make these penalty and rewards actions happen, the informative keywords in the context should first be identified to support these further adjustment to eventually get the uncertainty score for hallucination detection.

Experiments are conducted extensively on different LLMs mainly on wikiBio dataset. The experiment results well support the proposed approach. Ablation study is also given to present the boost performance of "penalty" and term statistics. More extensive experiments are provided in appendix.


**Reasons To Accept:**

(1) Uncertainty based hallucination calculation based on good observation of the hallucination phenomena.

(2) for the overconfident problem, propose penalty approach to mitigate the hallucination

(3) for underconfident problem, propose to adjust the score based on term statistics

(4) extenstive experiments to support the proposed approach

**Reasons To Reject:**

Not necessarily a strong negative point:

There is improvement space for the writing. Some suggestions are given in other section of the review.

**Reproducibility:**

4: Could mostly reproduce the results, but there may be some variation because of sample variance or minor variations in their interpretation of the protocol or method.

**Reviewer Confidence:**

3: Pretty sure, but there's a chance I missed something. Although I have a good feel for this area in general, I did not carefully check the paper's details, e.g., the math, experimental design, or novelty.

**Typos Grammar Style And Presentation Improvements:**

(1) abstract: hallucination is a well known research problem, it might be better to reduce the introduction of hallucination, while introduce more on the proposed approach. For example, what observation inspire authors to propose the approach in the paper? What are the major improvements on the benchmark evaluation? These would make the abstract more concrete and attractive to readers.

(2) Introduction: emphasize more on the findings and proposed approach. Some sentence could be revised in plain words. (e.g. Line 121 to Line 124)

---

> ### Author Rebuttal · Authors · 2023-08-29
>
> Thank you for your thoughtful review of our paper. We appreciate your positive feedback on our approach and findings. Below, we address your comments and indicate how we plan to revise the relevant sections in our paper:
>
> (1) We acknowledge your suggestions for revisions to the abstract. We will make the abstract more concrete and engaging in the final version by reducing the introduction to the hallucination and adding more about the motivation and underlying ideas of our approach.
>
> (2) We agree with your point. We will rephrase the sentences (lines 121\~124) in the final version to ensure that the message is conveyed concisely and effectively.
>
> Thank you again for your suggested changes and we will ensure that our final version will effectively address the concerns you've raised.

---

### Official Review · Reviewer_4M4D · 2023-08-07

**Soundness:** 3

**Excitement:**

3: Ambivalent: It has merits (e.g., it reports state-of-the-art results, the idea is nice), but there are key weaknesses (e.g., it describes incremental work), and it can significantly benefit from another round of revision. However, I won't object to accepting it if my co-reviewers champion it.

**Paper Topic And Main Contributions:**

Existing works for detecting hallucinations in LLMs either relies on external knowledge for reference retrieval or require multiple responses to be sampled from LLM for consistency verification, making these methods costly and inefficient. This paper proposes a novel reference-free, uncertainty-based method for detecting hallucinations in LLMs. Our approach imitates human focus in factuality checking from three aspects: 1) focus on the informative keywords; 2) focus on the unreliable tokens in history context; and 3) focus on the token properties.

**Questions For The Authors:**

(1) Are the results by LLaMA variants (without focus) reported in the paper (in case I did not see the results)?

(2) Is it possible to use some benchmarks as detailed in Section 4.2 in

Zha, Yuheng, et al. "AlignScore: Evaluating Factual Consistency with a Unified Alignment Function." arXiv preprint arXiv:2305.16739 (2023).?

(3) Can the proposed approach also work for some other model backbones?

**Reasons To Accept:**

(1) The paper studies an important and emerging problem. The problem setting is well motivated.

(2) The proposed solution has good intuitions and easy to follow.

(3) The proposed approach has strong empirical performances.

**Reasons To Reject:**

(1) More details can be included in the experiments. For example, in Table 1, LLaMA variants without focus can also be included. LLaMA-30B focus and LLaMA-65B focus clearly outperform the baselines. However, without showing the results by LLaMA variants (without focus), it is unclear whether the improvement is from (i) the difference between LLaMA and GPT or (ii) from the advantages by the proposed focus approach.

(2) I am not very sure about ‘We evaluated our proposed method on WikiBio GPT-3 dataset (Manakul et al., 2023), which, to the best of our knowledge, is the only publicly accessible dataset for LLM hallucination’? There could be multiple benchmarks, for example, as detailed in Section 4.2 in

Zha, Yuheng, et al. "AlignScore: Evaluating Factual Consistency with a Unified Alignment Function." arXiv preprint arXiv:2305.16739 (2023).

(3) To show the proposed approach is fundamental and generic, the paper may also consider to show the improvement by another model backbone (e.g., falcon).

(3) (Minor) More baselines can be added. Although it is understandable that the proposed method is not directly comparable to the LLMs, which either relies on external knowledge for reference retrieval or require multiple responses to be sampled from LLM for consistency verification. It is still interesting to show some baselines results, to understand the usability of the developed approach.

(5) (Minor) In Figure 1's caption, it is easy to understand ‘1) focus on the informative keywords’. However, it is difficult to find related illustrations for ‘2) focus on the preceding words; 3) focus on the token properties.’ in the figure.

**Reproducibility:**

3: Could reproduce the results with some difficulty. The settings of parameters are underspecified or subjectively determined; the training/evaluation data are not widely available.

**Reviewer Confidence:**

3: Pretty sure, but there's a chance I missed something. Although I have a good feel for this area in general, I did not carefully check the paper's details, e.g., the math, experimental design, or novelty.

---

> ### Author Rebuttal · Authors · 2023-08-29
>
> Thank you for raising these questions. We will explain each of your concerns below:
>
> *[Reasons to reject 1, Question 1]*
>
> Yes, the results by LLaMA variants without focus are reported in the ablation study of our paper. We present the ablation study of LLaMA-30b in Table 2 of Section 4.3 (lines 509\~512). And for all the 19 proxy models, the ablation study is presented in Table 9 to Table 26 of Appendix H. In the first row of each ablation study table, we presented the result of using loss and entropy of the corresponding model as a token-level hallucination score, **which is the result without focus as you mentioned**. From the results we can observe **incremental effectiveness of our proposed method with each focus trick incorporated, compared to the first row where no focus trick is applied**. We will modify the table captions and relevant content to make it easier to understand our ablation study settings.
>
> *[Reasons to reject 2, Question 2]*
>
> Yes, **we also conducted experiments on XSumFaith and FRANK datasets**, which are described in the dataset introduction section (Section 4.1 lines 393\~399) and **the results are presented in Appendix A**. It's worth noting that **these two datasets are derived from the SummaC benchmark**, which is detailed in the paper you referenced (Zha, Yuheng, et al. "AlignScore: Evaluating Factual Consistency with a Unified Alignment Function." arXiv preprint arXiv:2305.16739 (2023)). Since our paper primarily focuses on hallucination detection in **LLM** (Section 4.1, lines 399\~402) whose hallucination patterns tend to be divergent from those in small models or generated by humans (Sections 2.2, lines 171~177), we did not include these results in the main text. In the final version, we will consider moving these results to the main text to make them more visible to the readers.
>
> *[Reasons to reject 3, Question 3]*
>
> Yes, **our experiments were conducted on 19 diverse backbone models (Section 4.1, lines 445\~449), including falcon-7b and falcon-40b**. Due to space limitations, we only presented results of the LLaMA family in the main text (Section 4.2, lines 462\~465). **The full results for all 19 different backbone models, along with corresponding ablation study results, are included in Appendix H**. The ablation study results of all the 19 models demonstrate that **our method significantly enhances the hallucination detection capabilities of these backbone models compared to not using any focus tricks**. In the final version we will add more textual notes in section 4.2 to avoid misleading the readers that we only conducted the experiments on the LLaMA family.
>
> *[Reasons to reject 4]*
>
> Given that the current state of research on hallucination detection in **LLM** is still exploratory, the availability of viable baselines is limited. In the last two months, we found several related works, and we will consider adding them in the camera-ready version for comparison.
>
> *[Reasons to reject 5]*
>
> To illustrate how we "focus on the preceding words", we used red arrows in Figure 1 to indicate how uncertainties from preceding tokens are propagated to tokens in the succeeding context. To demonstrate "focus on the token properties", a green \<DATE\> tag is positioned before '1992', showcasing our utilization of in-context learning to guide the model's attention towards the specified token type of the succeeding word. We are sorry for the confusion caused by the mismatch between the caption and figure, we will consider rephrasing the caption to make it easier for readers to follow our ideas in our final version.

---

### Meta-Review · Area_Chair_HvRM · 2023-09-16

**Recommendation:** 5

**Metareview:**

This paper addresses the hallucination problem with a method more efficient than previous attempts that require multiple samples or external knowledge. Their uncertainty based method is said to resemble human focus in focusing on keywords, uncertain tokens in the context, and token properties. The reviewers agree that the paper is sound and exciting and mainly suggest minor revisions to improve the writing. The reviewers note the intuitive approach is valuable and performs well across diverse experimental settings. It appears that the contributions of this paper will be valuable to the community.

---

### Decision · Program_Chairs · 2023-10-07

**Decision:**

Accept-Main

**Comment:**

This paper addresses the hallucination problem with a method more efficient than previous attempts that require multiple samples or external knowledge. Their uncertainty based method is said to resemble human focus in focusing on keywords, uncertain tokens in the context, and token properties. The reviewers agree that the paper is sound and exciting and mainly suggest minor revisions to improve the writing. The reviewers note the intuitive approach is valuable and performs well across diverse experimental settings. It appears that the contributions of this paper will be valuable to the community.